# Exosomal miRNAs in the Tumor Microenvironment of Multiple Myeloma

**DOI:** 10.3390/cells12071030

**Published:** 2023-03-28

**Authors:** Shamila D. Alipoor, Hong Chang

**Affiliations:** 1Department of Laboratory Medicine and Pathobiology, University of Toronto, Toronto, ON M5S 1A8, Canada; 2Department of Molecular Medicine, Institute of Medical Biotechnology, National Institute of Genetic Engineering and Biotechnology, Tehran P5X9+7F9, Iran; 3Department of Laboratory Hematology, Laboratory Medicine Program, University Health Network, Toronto, ON M5G 2M9, Canada

**Keywords:** multiple myeloma, exosome, exosomal miRNA, drug resistance

## Abstract

Multiple myeloma (MM) is a malignancy of plasma cells in the bone marrow and is characterized by the clonal proliferation of B-cells producing defective monoclonal immunoglobulins. Despite the latest developments in treatment, drug resistance remains one of the major challenges in the therapy of MM. The crosstalk between MM cells and other components within the bone marrow microenvironment (BME) is the major determinant of disease phenotypes. Exosomes have emerged as the critical drivers of this crosstalk by allowing the delivery of informational cargo comprising multiple components from miniature peptides to nucleic acids. Such material transfers have now been shown to perpetuate drug-resistance development and disease progression in MM. MicroRNAs(miRNAs) specifically play a crucial role in this communication considering their small size that allows them to be readily packed within the exosomes and widespread potency that impacts the developmental trajectory of the disease inside the tumor microenvironment (TME). In this review, we aim to provide an overview of the current understanding of the role of exosomal miRNAs in the epigenetic modifications inside the TME and its pathogenic influence on the developmental phenotypes and prognosis of MM.

## 1. Introduction

Multiple myeloma (MM) is the second most common hematological cancer [1]. This cancer is characterized by the clonal proliferation of malignant plasma cells within the bone marrow (BM), which results in the production of abnormal antibodies and end-organ damage manifested as renal impairment, anemia, or bone lesions [2]. The development of MM is considered a multistage process and almost all cases of MM are preceded by a pre-malignant stage termed monoclonal gammopathy of undetermined significance (MGUS). MGUS is an asymptomatic monoclonal plasma-cell proliferation characterized by moderately increased levels of a defective immunoglobulin. MGUS does not require treatment, but it as an important risk factor for the development of MM [3], considering that MM is an incurable disease. Although the current therapeutic strategies combined with the introduction of novel drugs have improved disease outcome, drug resistance is still a major challenge and most patients relapse and do not respond to chemotherapies [4].

Bone-marrow microenvironment (BME) has an important role in the progression of disease and/or developing drug resistance (DR) [5]. It has been shown that several molecular pathologies in BME predispose conditions for the survival and growth of the mutant plasma cells and transformation from an asymptomatic state to malignancy [5]. The crosstalk between the malignant cells and other components in this microenvironment plays a determinantal role in the fate of the disease. Therefore, understanding these interactions is the key point to elucidate the mechanisms of disease and drug resistance in MM. Among the several interactions within the BM, exosomes have emerged as the main players in cell-to-cell communication. Exosomes are small (30–150 nm) extracellular vesicles secreted by nearly all cell types and contain functional biological molecules originated from their parent cells. Exosomes shuttle important information between cells and are also decorated with surface signaling molecules which can trigger the cell-signaling pathways [6]. According to ExoCarta, a database of exosomal contents, a wide variety of complex compositions, including 9769 proteins, 1116 lipids, 3408 mRNA, and 2838 miRNA, has been identified in exosomes from multiple organisms [7]. In the tumor condition, exosomes can shuttle their tumorigenic cargo into the extracellular milieu and process the critical biological functions in the recipient cells, even at a distance. Hence, exosomes are considered the main players of the TME.

miRNAs, belonging to the class of small-noncoding RNAs, are important regulatory molecules that play a role in the tuning of the gene expression at the post-transcriptional level [8]. The miRNA cargo in the exosomes is currently attracting increasing attention in cancer biology due to its role in the reprogramming of the important pathways in TME [8]. In MM, miRNAs are important elements in the progression of disease, and their role in the regulation of BME remodeling, epigenetic modifications, immune regulation, and drug resistance is well documented [9,10,11,12]. Regarding their critical functions in the pathogenicity of MM, the shuttling of miRNAs by exosomes within BME can change the phenotype of recipient cells and direct them towards malignancy. There is some evidence that miRNAs are not randomly sorted but rather selectively packaged inside the exosomes during their biosynthesis process [13,14]. In addition, recent data indicate that exosomes are selectively taken up by the other cells in the TME. Exosome contents and cell-surface molecules, such as integrins, affect the selectivity to recipient cells [15]. The selective packaging and uptake of exosomal miRNAs strengthen their roles in the BME compared to those from the non-exosomal or intracellular sources. On the other hand, surface molecules on the exosomes can trigger signaling pathways in the recipient cells, which reinforce the function of exosomal miRNAs.

In this work, we aim to review the role of exosomal miRNAs inside the TME of myeloma and discuss how they affect MM progression and phenotypes.

## 2. Exosomal miRNAs and Epigenetic Modifications in BME Microenvironment: The Potential Roles in the Progression of MGUS to MM

Multiple myeloma is developed from the precursor conditions MGUS and smoldering multiple myeloma (SMM) in a process known as myelomagenesis. MGUS is the pre-malignant asymptomatic version that is present in approximately 5% of the general population above the age of 50 and mainly remains stable. However, it may progress to MM with a rate of 1% per year [16]. MGUS typically progresses to SMM in the first stage before developing malignancy. SMM has a much higher risk of progression to MM, with a risk of approximately 10% per year [16].

The BME microenvironment plays an important role in myelomagenesis. Along with the genetic and epigenetic backgrounds that predispose individuals to the development of MM, other alterations in the bone marrow microenvironment, including changes in cytokine profile and the presence of immune suppressive cells coupled with dampened immune surveillance, may change the phenotype in BME towards a pre-cancerous condition [5]. The profile of exosomal miRNAs plays an important role in the immune phenotype of BME. It has been reported that the exosomal miRNA profile differs between normal, MGUS, and SMM patients [17]. In a study by Manier et al., a higher expression of miR-107 and lower expression of miR-28, -32, -548a, -99a, -345, -125a, and -92a was observed in SMM compared to MGUS-derived serum exosomes [18]. MiR-99a, -92a, -28, -345, -125a, and -548 are reported to function as tumor-suppressor miRNAs. Specifically, miR-125a has been reported to reduce malignancy in MM cells by targeting the angiogenesis factors and deubiquitinase USP5 [19], while miR-28-5p controls cell proliferation and MYC activation in B-cell lymphomas [20]. Moreover, miR-28-5p also suppressed cell migration by targeting WSB2 in breast cancer cells and exerted multiple antitumor effects in renal-cell carcinoma by targeting RAP1B [21]. Therefore, these findings indicate that during the transition from MGUS to SMM, the profile of exosomal miRNAs changes towards an immune-suppressive and tumorigenic profile that may favor the condition in BME for disease progression [18] (Table 1, Figure 1).

miR-21 is another miRNA that is upregulated in the primary tumors in MM patients and plays a role in the transformation from MGUS to MM. The miR-21 promoter is strongly controlled by STAT3, which is activated by IL-6 [22]. IL-6 is a key player in the myelomagenesis mechanisms, and the IL-6/STAT3 pathway via miR-21 contributes to the development of malignancy in the initial stage of the disease [22,23]. However, exosomal miR-21 is secreted by different cell types in the TME, and uptake the of exosomal miR-21 by normal PCs may trigger the tumorigenic signaling pathways in these cells independent from the IL-6 [22,23]. On the other hand, miR-21 can indirectly induce the expression of STAT3 in a positive feedback loop by targeting its inhibitors (PIAS3) and promote angiogenesis and malignancy in normal cells [24,25]. Furthermore, miR-21 regulates the synthesis process of piRNAs that have a role in the epigenetics mechanisms and preserving genetic stability in germlines. piRNAs recruit DNMTs into primary CD138^+^ MM cells and enhance DNA methylation and silencing of the tumor suppressors and miRNAs genes [26,27]. Therefore, the uptake of the exosomal miR-21 may contribute to the methylation and silencing of miRNA genes in the recipient plasma cells, as it has been shown that the promoter of miRNA-34a, -15, and -16 genes are highly methylated in malignant plasma cells [28].

In normal conditions, these exosomal miRNAs contribute to the stability and expression of Sirtuin 1 (Sirt1) [29,30], and their downregulation in the TME leads to the overexpression of Sirt1 and MM progression [31]. Sirt1 is a NAD-dependent deacetylase belonging to the family of sirtuins that are involved in a myriad of cellular processes from aging to cancer. The downregulation of these miRNAs also promotes the overexpression of the enhancer of zeste homolog 2 (EZH2), which subsequently changes the gene-expression profile in BME [32]. EZH2 is an essential epigenetic modifier in MM [33]. It is an enzymatic catalytic subunit of Polycomb Repressive Complex 2 (PRC2) and mediates the trimethylation of Lys-27 in histone 3 (H3K27me3) and the silencing of its target genes [34]. The high levels of EZH2 and the increased number of silenced genes via H3K27me3 were associated with the end stage of the disease [35]. Furthermore, an increase in EZH2 expression along with other core subunits of the PRC2 complex, including SUZ12 and EED, was reported during MGUS and SMM progression to MM [36].

The BME also can directly change the expression of DNMTs and HDACs using the regulatory role of exosomal miRNAs [27]. Exosomal miR-29a-3p released by TAMs regulates DNA methylation by targeting DNMT1, DNMT3b, and DNMT-3a. In addition, exosomal miR-9 transferred by MM tumor cells targets HDACs [27] and Sirt1 [37]. Exosomal miR-29b in BME targets HDACs [38,39], DNMT3A, and DNMT3B [40]. Interestingly, the increased expression of HDACs and the methylation status of the tumor-suppressor genes in MM samples were associated with the advanced stages of the disease and the overall survival of the patients, respectively [41]. Therefore, targeting epigenetic modifiers may be an efficient strategy for the treatment and management of MM progression.

On the other hand, lifestyle may also influence the myelomagenesis process. Exercise, diet, pollution, and morbidities, as well as smoking and addiction, may change the profile of exosomal miRNAs [42]. Diet and nutrition may affect MM progression by modulating the epigenetic patterns. In a cohort study by Chang et al., obesity was reported as a risk factor in the progression of MGUS to MM [43]. In metabolic syndromes, caloric restriction (CR) attenuates inflammation and improves metabolic syndrome features by altering DNA methylation and histone modification [44]. In addition, the consumption of fatty acids (FA) can alter the expression of miRNAs and epigenetic patterns [45]. Furthermore, polyphenols in the daily diet can modify histones and suppress DNMTs toward acquiring an antitumor phenotype. For example, green tea modified the expression of histone-acetyltransferase inhibitors and miRNA expressions [46]. Interestingly, it has been reported that physical activity altered the exosomal miRNA profile, and the changed miRNAs were involved in chronic diseases, including cancer and infections [47]. Smoking and addictions impact the chromatin structure through the reprogramming of histone modifications [48]. Furthermore, exposure to environmental pollutants, including air pollution, can alter the miRNA expression and interfere with epigenetic factors and promote histone modifications and DNA methylation [49]. These harmful changes have a risk of transferring across generations, resulting in transgenerational inheritance and increased individual susceptibility to developing MM malignancy [50].

**Table 1 cells-12-01030-t001:** The potential role of exosomal miRNAs in the transition from MGUS to SMM [18].

miRNA	Expression	Potential Role	References
miR-107	Upregulated	Targeting HF-1β	[51]
miR-32	Downregulated	Modulation of P53 and MDM2	[52]
miR-223	Upregulated	OncomiR, immune-cell differentiation	[53,54]
miR-28	Downregulated	Cell proliferation and MYC activation	[20]
miR-345	Downregulated	Tumor suppressor	[55]
miR-99a	Downregulated	Tumor suppressor	[56,57]
miR-125a	Downregulated	Targeting the angiogenesis factors	[19]

## 3. BM Tumor Microenvironment and the Critical Reactions in Multiple Myeloma: Possible Roles of Exosomal miRNAs

The BME consists of cellular and noncellular compartments. The cellular compartment contains the stromal cells, endothelial cells, immune cells, osteoclasts, and osteoblasts, while the noncellular compartment contains cytokines, growth factors, chemokines, and the extracellular matrix (ECM). The different components in the cellular portion of the BM milieu form a composite structure with ECM proteins that shelter the malignant plasma cells (MM tumor cells) to create a specialized niche in which MM cells are protected from immune surveillance [58].

Within this niche, which can be referred to as the TME, MM cells induce the phenotypic alteration of BME towards an immune-suppressive environment by sending different signals through direct contact with stromal cells, the stimulation of cytokines and supportive soluble factors, and the production of exosomes. These signals trigger further responses to promote MM tumors and alter the ECM [59,60].

Tumor-associated macrophages (TAMs), cancer-associated fibroblasts (CAFs), myeloid-derived suppressor cells (MDSCs), and endothelial cells (ECs) are the major components in the TME and are considered “ obligate partners” for tumor progression [61]. The phenotype of tumor cells is strongly dependent on the crosstalk between these components within the TME. Exosomes regulate these interactions by mediating the bidirectional crosstalks through shuttling functional molecules, including exosomal miRNAs [62]. The delivery of the exosomal miRNAs in the TME leads to changing the gene-expression pattern in the recipient cells and reshaping the TME. It has been shown that in MM, tumor cells educate BM cells to produce exosomes with a higher level of oncogenic miRNA contents, which reshape the TME by the induction of vascular disruption and leakiness, increasing angiogenesis and promoting an immune-suppressive environment [63].

**Bone marrow stromal cell (BMSC)**-derived exosomes have significant effects on the phenotypes of the TME in MM. Exosomes from the BM microenvironment induce transcriptional dysregulations in the tumor cells [64]. Roccaro et al. observed that BMSC-derived exosomes from MM patients enhanced tumor growth, while those from healthy donors had antitumor effects and suppressed tumor proliferation and survival [65]. Additionally, the profile of BMSC-derived exosomal miRNAs in MM patients and healthy subjects are different [65]. A group of miRNAs, whose deficiency in MM tumors promotes pathogenicity and drug resistance in these cells, are downregulated in the BMSc-derived exosomes from MM patients (MM- BMSC exosomes). Interestingly, these groups of miRNAs are packaged in healthy BMSC exosomes with a higher concentration (GSE78865; GSE39571). miR-15a and -16 are enriched in BMSC-derived exosomes from healthy subjects, while they are downregulated in those from MM patients [65]. miR-15a and -16 exert tumor-suppressive activity by targeting AKT serine/threonine protein kinase (AKT3) and are downregulated in MM tumor cells [66]. Furthermore, the expression level of miR-152 in healthy BMSC-derived exosomes is higher than it is in MM-BMSC exosomes (GSE78865; GSE39571). This miRNA is also downregulated in MM cells [67]. These findings suggest that the BMSCs and tumor cells share similar genetic profiles and have strong crosstalk within the TME. Interestingly, co-culturing normal human BMSC with the conditioned medium of MM cells led to the changing in the miRNA profile of BMSCs, and upregulation of oncomiRs, including miR-146a and miR-483-5p [68]. On the other hand, MM-tumor-derived exosomes increased the proliferation of MSCs by shuttling miR-146a and miR-21 to these cells [61]. These exosomes also induce IL-6 production by MSCs, which favor the conditions for MM cell growth [61].

The uptake of MM exosomes by BMSC also enhanced the production of cytokines and chemokines and promoted cell viability and migration in these cells by targeting apoptosis-related proteins and TIMPs, respectively [68]. It also has been reported that BMSCs regulate the expression of miR-30 in MM cells, and co-culturing MM cells with BMSCs reduced the amount of miR-30 in MM cells, and subsequently increased MM cell proliferation by promoting the Wnt signaling pathway [69]. Furthermore, BMSC-derived exosomes shuttle miR-182 to the MM-tumor cells, which can contribute to cell-adhesion-mediated drug resistance (CMDR) in MM cells via targeting PDCD [70]. In the process of CAM-DR, the adhesion of myeloma cells to fibronectin (FN) via its surface integrinβ1 induces the expression of miR-182, which suppresses the expression of human programmed cell death 4 (PDCD4) and triggers downstream drug-resistance mechanisms. This process leads to the increased expression of cell-cycle-inhibitory protein p27Kip1 and G1 arrest and the inhibition of cyclin-A- and cyclin-E-associated kinase activity [71,72]. The uptake the exosomal miR-182 may trigger drug-resistance mechanisms in the recipient cells independent of direct cell–cell contact. The overexpression of miR-182 was also associated with drug resistance in lymphatic malignancies [73]. In addition, it was reported that in colorectal cancer cells, the high expression of miR-182 interfered with cell growth in the G0/G1 phase and suppressed the apoptosis pathway [74]. Furthermore, miR-182 promoted invasion and metastasis by targeting the notch pathway through the NF-kB-miR-182-HES1 axis and decreasing the expression of cell-adhesion molecules in gallbladder cancer [75,76].

It has been reported that miRNA-181 is overexpressed in CD138^+^ PCs from MM patients [9,77]. miR-181 is released to BME by BMSC-derived exosomes and may be uptaken by PCs [78]. This miRNA epigenetically regulates the tumor-suppressor activity of p53 and Sirtuin 1 (SIRT1) [9,78]. Sirt1 is a histone deacetylase (HDAC) and plays an important role in epigenomic profiling in MM by modulating the expression of P53 and miRNA-34a, and its targeting is suggested as a treatment strategy for MM [31].

Collectively, these finding indicate that the crosstalk between BMSCs and the TME has an important role in the phenotype of disease, and changing this profile may contribute to acquiring an aggressive phenotype or resistance to therapy.

**Tumor-associated macrophages (TAMs)** are another population of immune cells that create an immunosuppressive tumor microenvironment (TME) and play crucial role in MM progression [79]. TAMs are mostly populated in the vicinity of CAFs, which suggests a tight interaction between these cells [80]. CAFs recruit monocytes to the TME and promote their polarization into TAMs, which helps the formation of an immunosuppressive microenvironment around MM cells [81]. The increased infiltration of macrophages to the tumor site is a major contributor to tumor progression in MM and is correlated with the initial response to bortezomib therapy [82].

Similar to the other cells in the TME, TAMs release exosomal miRNAs and affect the TME. Exosomal miR-365 released by TAMs induces gemcitabine (a cytidine analog) resistance in pancreatic cancer cells. This miRNA promotes pyrimidine metabolism by the upregulation of the cytidine deaminases (CDA), disabling gemcitabine and providing essential nucleotides for the proliferation of the tumor cells [83,84]. However, the intracellular source of this miRNA has been identified with a protective role against tumor progression. Intracellular miR-365 is overexpressed in myeloma tissues and induces apoptosis and inhibits metastasis by modulating homeobox A9 (HOXA9) [85]. Likewise, in hepatocellular carcinoma, intracellular miR-365 inhibited the cell growth and metastasis of tumor cells by targeting ADAM10 [86]. miR-501-3p is enriched in TAM-derived exosomes and modulates transforming growth factor beta (TGF-beta) signaling [87]. TGF-β is a strong regulator of normal B-cell development [88]. In MM, TGF-β is secreted at higher levels from both tumor and bone marrow stromal cells and promotes myelomagenesis, drug resistance, and osteoblast dysfunction [89]. In the hypoxic condition, in different cancers, it has been reported that TAMs release exosomal miR-223, which reshapes the TME and induces drug resistance by activating the PTEN/PI3K/AKT pathway [90,91,92,93]. Interestingly, it has been reported that the serum-exosomal miR-223 is increased in MM patients and can be considered a potential diagnosis biomarker [94].

In addition, lncRNA SBF2-AS1 in TAM exosomes promoted the expression of the X-linked inhibitor of apoptosis protein (XIAP) and tumor progression in prostate cancer cells [95]. XIAP is a robust endogenous inhibitor of caspases and is highly expressed in myeloma cells. The inhibition of XIAP production promoted drug sensitivity and impeded tumor formation in an MM model of NOD/SCID mice [96]. TAMs also have a strong crosstalk with CD4^+^ T cells in the TME. miR-29a-3p and miR-21-5p are enriched in TAM-derived exosomes and are able to reprogram CD4+ T-cells and induce an imbalance in the Treg/Th17 ratio, which promotes an immune-suppressive microenvironment around tumor cells [97].

**Cancer-associated fibroblasts (CAFs)** are a subgroup of fibroblasts inside the tumor stroma that have a crucial role in promoting tumor growth. Normal fibroblasts mainly take part in tissue homeostasis, wound healing, and the regulation of inflammation, but in tumor conditions, they may be activated, and promote a pro-tumorigenic feature. These activated fibroblasts, called cancer-associated fibroblasts (CAFs) or pretumor fibroblasts, fail to be inactive and revert to a normal phenotype in tumor tissue [98]. The presence of CAFs is considered the major component in the TME of MM [99]. Previous studies have shown a higher proportion of CAFs in the BME of patients with active MM compared to those in remission or with MGUS [100].

There is a strong crosstalk between CAFs and the TME of myeloma, mediated by the exosomal miRNAs (Table 2, Figure 2). Tumor plasma cells transfer exosomal miR-105 and miR-122 to CAFs, which induce MYC activation and metabolic reprogramming in these cells, leading to the release of metabolites in the TME to fuel cancer cells [101,102]. Exosomal miR-27b-3p and miR-214-3p released from the MM tumors also enhanced proliferation and apoptosis resistance in myeloma fibroblasts and promoted the transition from MGUS to myeloma [103]. On the other hand, CAFs secrete exosomes that contain specific tumorigenic and chemoresistance-promoting microRNAs, effectively modulating the TME and promoting tumor progression and drug resistance. It has been reported that CAF exosomes transfer the miR-181 family [104], which is enhanced in tumor plasma cells, especially in drug-resistant MM cells, and control the expression of P53 [12], block apoptosis, and promote MM cell proliferation [12,105,106]. In addition, these exosomes shuttle miR-20a inside the TME. This miRNA targets the PTEN/PI3K/AKT signaling pathway and promotes proliferation, migration, and apoptosis in MM cells [107]. CAF-derived exosomal miRNAs exert similar effects on promoting malignancy in the TME of solid tumors. In breast cancer, tumor cells educate normal fibroblasts to acquire a CAF phenotype by transferring exosomal miR-9 [108]. CAF-derived exosomal miR-20a enhanced chemoresistance in non-small-cell lung cancer (NSCLC) [109] and targeted the Wnt/β-catenin signaling pathway in HCC cells, which led to the suppression of LIMA1 and tumor progression [109]. In pancreatic cancer, miR-146a levels along with the mRNA of Snail were highly increased in CAFs exosomes during GEM treatment. Snail is the upstream transcription factor of miR-146a [110]. It is reported that the expression of Snail in bortezomib-resistant MM cells was significantly higher than that in the susceptible ones [111]. In MM, Snail induces drug resistance in MM cells by upregulating MDR1 genes and downregulating P53 [111]. The shuttling of miR-146a and Snail by CAF-derived exosomes within the TME may play a role in developing drug resistance in MM cells.

CAFs also release exosomal miR-22 that induces resistance to tamoxifen by targeting ER***α*** and PTEN in breast cancer [112]. Interestingly, the non-exosomal source (intracellular) of miR-22 exerted anti-drug resistance effects in breast cancer cells, improved radiosensitivity in these cells by targeting Sirt1, and suppressed tumorigenesis [113]. In MM cells, intracellular miR-22 improved the response to immunomodulatory imide drugs (IMiDs) by targeting MYC addiction in MM cells [114]. MYC has a major role in the development of resistance to anti-MM drugs and the progression of the disease by the regulation of the miRNAs [114]. On the other hand, intracellular miR-22 in MM cells targets the LIG3 protein, which leads to increased DNA damage and death in MM cells. Since the high genomic instability is the main character of MM, the hyperactivation of DNA ligase III (LIG3) is crucial for the survival of the neoplastic plasma cells. It has been observed that the expression of LIG3 mRNA in MM patients enhanced in the more advanced stage of the disease and was correlated with shorter survival [115].

The packaging of miR-22 inside the exosomes is mediated by SFRS1, a protein that regulates selective exosome miRNA enrichment in human tumor cells [116]. The exosomal surface CD63 in the *CD63^+^ CAF*-derived exosomes induces STAT3 activation in the recipient cells, which may be responsible for the different role of exosomal miR-22 in the induction of drug resistance in tumor cells compared to its intracellular source [112].

CAF-released exosomes also contain miR-21, which is a well-known oncomiR in the myeloma microenvironment [117] and induces anchorage-independent cell growth, stemness feature, and EMT in breast tumor cells [118]. CAF-derived exosomal miR-21 is responsible for generating myeloid- derived suppressor cells (MDSCs) via activating STAT3 signaling [119].

**Myeloid-derived Suppressor Cells (MDSCs)** are a functionally heterogeneous population of immune cells that expand during the pathological condition, including cancers. They have a crucial role in developing an immunosuppressive niche in the MM tumor microenvironment by the secretion of suppressor cytokines. MDSCs promote T-cell anergy and T-reg development in the MM microenvironment [120] and inhibit T-cell function by releasing arginase and inducing nitric oxide synthase [121]. It has been reported that the uptake of BMSC-derived exosomes by MDSCs triggers the STAT3 and STAT1 pathways and promotes myeloid leukemia cell differentiation protein (MCL-1) expression, which favors the survival and expansion of MDSCs in the myeloma TME [122].

It also has been shown that exosomal miRNAs in the TME of solid cancers significantly mediate the activity of MDSCs.

In breast cancer, the exosomal miR-9 and miR-181a released by the tumor cells promote the proliferation and development of MDSC by targeting SOCS3 and the activation of the JAK/STAT pathway [123]. The miR-21 in TEXs promotes the activation of MDSCs via phosphorylation and triggering the STAT3. miR-21 also may reinforce the immunosuppressive function of MDSCs through targeting the miR-21/PTEN/PD-L1 axis [124].

miR-10a in the glioma hypoxia-stimulated TEXs modulated MDSC activity by targeting the miR-10a/Rora/IkBa/NFkB pathways which applied a more aggressive suppression on CD8+ T cells [125]. Furthermore, the transfer of miR-29a and miR-92a from the TEXs to MSDCs also showed similar effects via triggering the miR-29a/H.

MGB1 and miR-92a/Prkar1a pathways in glioma cancer cells [126]. Whether this function also occurs in the BM hematopoietic niche and affects TME immune properties during myeloma development remains to be elucidated. Next-generation sequencing has shown that MDSC-derived exosomes are enriched with immune-suppressive miRNAs, which mainly induce the expression of the suppressive cytokines and also target apoptosis-related proteins, including Fas and Fas ligands [127]. miR-155 is one of the MDSC-released exosomal miRNAs, which increases the expression of IL-10 in the TME and promotes the differentiation of macrophages towards the M2 phenotype [128]. miR-155 directly targets Jumonji and At-rich interaction domain-containing 2 (JARID2), which is a transcriptional repressor for inflammatory cytokines and increases IL-10 expression in the regulatory B-cells (Bregs) [129]. Regulatory B-cells (Bregs) are a small subgroup of B-cells with immunosuppressive function. In MM, Bregs suppress the anti-MM activity of NK cells and promote an immunosuppressive microenvironment [130]. It is reported that the population and activation of Bregs are increased at the very beginning stage of MM [131] and during the transition from MGUS to MM [132]. It also has been observed that the proportion of Bregs in MM patients is closely correlated with treatment efficacy and prognosis [133,134].

Targeting Jarid2 by miR-155 also regulates T-cell differentiation and cytokine expression in Th17 cells [135]. TH17 cells have an important role in the TME of myeloma, and it is reported that the expression of IL-17 by TH17 cells is increased in MM patients, which attenuates immune function and promotes myeloma cell growth [136,137]. These findings highlight how changing the exosomal miRNA profile can affect the phenotype and function of the immune cells within the TME and favor the condition for developing malignancy.

**Endothelial cells(EC)** are the major components of the BME and regulate this microenvironment via the production of cytokines and exosomes. Crosstalk between BM-endothelial cells and the TME via exosomal miRNAs plays a determinant role in the progression of the disease. The treatment of MM tumor cells with bortezomib changed the profile of their microvesicles with less angiogenetic factors, which had reduced effects on tube formation by human-umbilical-vein endothelial cells (HUVEC) [138]. In addition, serum exosomes obtained from the MM patients induced a stronger nuclear factor kappa B (NF-kappa B) translocation and proliferation effect in ECs compared with MGUS serum exosomes. These exosomes contained a higher level of proto-oncogene c-src kinase in MM patients, which promoted cell proliferation and survival and induced chemoresistance in MM via NFκB pathways [139].

Furthermore, it was reported that MM exosomes delivered piRNA-823 to the endothelial cells and induced the expression of tumorigenic factors, including VEGF, IL-6, and ICAM-1, and inhibited apoptosis [27,140]. The synthesis pathway of piRNAs in ECs also may be induced by TAM-released exosomal miR-21 [140]. piRNA-823, as a member of the piRNA family, is the main regulator in the pathogenesis of MM and promotes tumor-cell proliferation in this disease by the regulation of the epigenetic gene silencing. The increased levels of piRNA-823 were associated with late stages and poor prognosis of the disease [26,141].

MM exosomes also transfer pro-angiogenic-related miRNAs to ECs, which directly induced new tube formation via STAT3 phosphorylation. Exosomes produced by the hypoxia MM cells were enriched with miR-135b that targeted Factor-inhibiting HIF-1 (FIH-1) in the recipient endothelial cells (ECs) and enhanced angiogenesis [142].

Interestingly C6-ceramide treatment inhibited the induction of angiogenesis by MM exosomes. Ceramide is a type of sphingolipid that plays role in exosomal secretion [143]. In MM, ceramide pathways have an important role in the modulation of MM survival via exosomal miRNA regulated mechanisms. It has been shown that in MM cell lines, C6-ceramid increased the exosomal level of antitumor miRNAs, including miR-29b, miR-202, and miR-15a/16 [144], which simultaneously exerted anti-angiogenesis and antitumorigenesis effects by targeting the Akt pathways [143].

In addition, BMSC-derived exosomes from the healthy and young subjects inhibited angiogenesis by transferring miR-340 and interfering with the Met signaling pathways in an in vivo model of MM with hypoxic bone marrow [145]. More understanding of the role of exosomal miRNAs in mediating the function of ECs could offer new opportunities for managing angiogenesis and disease progression via targeting these pathways.

**Table 2 cells-12-01030-t002:** The role of exosomal miRNAs and their crosstalk in the development of MM.

Cells	Exosomal miRNAs	Potential Function
TAM	miR-223 [94], miR-365 [83,84], miR-501 [87], miR-29a [97]	Proliferation of tumor cells, apoptosis inhibition, and drug resistance, promoting the immunosuppressive TME
CAF	miR-20a [107], miR-181 [104], miR-22 [112], miR-21 [117]	Tumor growth and drug resistance
MDSCs	miR-155 [128]	Developing the immunosuppressive niche
Endothelial cells	miR-214 [146]	Promoting angiogenesis
MSC	miR-152 [67], miR-30c [69], miR-182 [70], miR-181 [78]	Tumor growth and drug resistance

CAF: cancer-associated fibroblast; TAM: tumor-associated macrophages; MDSCs: myeloid-derived suppressor cells; MSCs: mesenchymal stem cells.

## 4. Potential Therapeutic Strategies Targeting the Exosomes in MM

Drug resistance is the main challenge in management in a broad range of cancers, especially in MM. In the process of myelomagenesis, the bone marrow microenvironment provides an intricate network of cellular communication to favor the tumor niche formation [147]. Targeting this intracellular network provides new opportunities for the therapy of the disease. Regarding the importance of exosomal miRNAs in the communications within the BME in developing drug resistance and the progress of the disease, they are considered as desirable therapeutic targets in MM treatment interventions [147]. Exosome-based strategies in cancer therapy or diagnosis have currently achieved a high level of interest, crucially focusing upon (1) inhibiting exosome-mediated crosstalks by reducing the production of exosomes and/or blocking their uptake, (2) their usage as therapeutic tools for the delivery of drugs, and (3) their role as potential biomarkers for early diagnosis, prognosis, or response to therapy.

### 4.1. Exosome Inhibitors

Regarding the role of exosome crosstalks in cancer progression, the inhibition of exosome release/uptake has been an attractive target for cancer treatments. In a study by Zheng et al., pharmacological inhibitors, including heparin, wortmannin, dynasore, and omeprazole have been used to block different uptake routes of specific exosomes and inhibit the tumor progression [148]. Furthermore, blocking exosome endocytosis by chemical inhibitors sensitized MM tumors to bortezomib treatment [149]. Many efforts have been made so far to design efficient exosome inhibitors. Most of these inhibitors are derived from synthetic compounds and are currently considered drugs.

SST0001 is a chemically modified heparin with anti-heparinase activity and has shown the potential to suppress exosome secretion in MM cells [150]. Heparinase activity in tumor cells enhances exosome secretion and alters the tumor exosomal cargo with tumorigenic factors which promote tumor progression [150]. The inhibition of the heparinase activity by SST0001 suppressed MM progression in vivo, even when confronted with an aggressively growing tumor within the human bone [151].

GW4869, another inhibitor of exosome release, is a neutral sphingomyelinase and inhibits exosome release from the plasma membrane and has shown cytotoxic effects on the MM cells [152] Interestingly, GW4869 reduced osteolysis and led to a reduction in tumor growth and angiogenesis in 5TGM1 mice [153].

Furthermore, the inhibition of the MM-exosome uptake by BMSCs using the endocytosis inhibitors suppressed exosome-induced changes in these cells and inhibited tumor-cell survival and proliferation [148].

Interestingly, it has been shown that some of the anticancer and anti-angiogenesis natural compounds can reduce exosome secretion and alter exosomal miRNA contents [154]. D Rhamnose β-hederin (DRβ-H), an active component isolated from a natural Chinese plant, reduced the secretion of exosomes and subsequently inhibited the growth and promoted the apoptosis of breast cancer cells by decreasing the level of tumorigenic encapsulated miRNAs [155,156]. DRβ-H also reversed the chemoresistance of breast cancer cells by reducing the secretion of exosomes from drug-resistant cells as well as the regulation of exosomal miRNA expression [157]. In addition, Shikonin (SK) inhibited exosome release and reduced the exosomal miR-128 level [154].

Regarding the potential of these components in the regulation of exosome release as well as the exosomal miRNA content, they may be practical in controlling disease progression or therapy resistance in MM patients. DHA could alter exosome secretion and exosomal miRNA contents and inhibit tumor growth [140]. In addition, the treatment of breast cancer cells with Epigallocatechin gallate (EGCG) increased miR-16 exosomal encapsulation and inhibited macrophage infiltration and M2 polarization [158]. Regarding the important role of miR-16 in MM tumor cells, it would be worth to evaluate the role of these neutral components in the inhibition of exosomal crosstalk and MM progression.

### 4.2. Exosomal miRNA Delivery as Therapeutic Strategy

There is some evidence for using engineered exosomes for the specific targeting of tumor cells and delivery of synthetic miRNA to inhibit the tumor cell growth. miR-21 is one of the main exosomal miRNAs in BME during myelomagenesis, and its targeting has been reported with therapeutic effects [117,159]. Ibrutinib suppresses the expression of miR-21 expression in MM cells by inhibiting nuclear factor-κB and STAT3 signaling pathways and was suggested as a promising potential treatment for this disease [160]. In addition, the attenuation of miR-21 with the NL101 drug suppressed the growth of B-cell lymphoma by targeting the c-Myc/Mxd1 loop [159].

The glioblastoma-targeted delivery of antisense miR-21 using engineered exosomes was performed by Kim et al. [161]. The intravenous injection of this synthetic drug into glioblastoma rat models attenuated the expression of miR-21 and promoted the expression of PDCD4 and PTEN in animals and suppressed tumor growth [161].

In addition, there is a great deal of evidence on the therapeutic effects of the artificial inhibition/overexpression of the miRNAs in MM to prevent tumor progression (Table 3).

The overexpression of miR-29 in the BME impaired the differentiation and activation of osteoclasts [168]. In addition, miR-29b modified Th1 differentiation by targeting interferon-ɣ and exerted potent antimyeloma activity [169].

Furthermore, the inhibition of IRF4 by synthetic miR-125b-5p induced the antitumor activity against MM tumors in vitro and in vivo. IRF4 plays a crucial role in the biology of Treg and Th17 cells [144]. On the other hand, Di Martino et al. reported evidence of a tumor-suppressor function of miR-34a in MM. They have shown that the synthetic miR-34a mimics inhibit growth and promote the apoptosis of MM cells by the suppression of CDK6, BCL-2, and NOTCH1 expression [166].

Although many studies have confirmed the usage of miRNAs as promising therapeutic agents for MM, no miRNA molecules have been conducted in clinical trials for MM patients. There are some challenges to be overcome to achieve an efficient and optimized therapy method, including maintenance of the stability of miRNA molecules or the efficiency of delivery methods.

For solving these problems, several methods have been tried, for instance, the conjunction of miRNAs with liposome nanoparticles. A liposomal formulation of miR-34a (MRX34) is in phase 1 clinical trials for the treatment of patients with solid tumors [170]. MRX34 is a miRNA drug that contains a double-stranded miR-34a mimic, which is encapsulated in a liposomal nanoparticle [170]. The antitumor activity of miR-34a is important in hematologic malignancies, and it is downregulated in MM patients [171,172]. Therefore, the miRNA drug MRX34 may offer new therapeutic approaches for MM.

In addition, miR-16 mimics are now in a Phase I clinical trial for patients with malignant pleural mesothelioma. TargomiRs are minicells loaded with miR-16-based mimic miRNA and targeted to EGFR that are designed to improve the loss of the miR-15 and miR-16 family miRNAs in tumor cells [173]. Regarding the role of miR-15 and -16 deficiency in the progression of MM malignancy and drug resistance, this drug may be practical for this disease.

### 4.3. Potential of Exosomal miRNAs as Biomarkers

Regarding the importance of monitoring the patients in the primary stage of MM for developing malignancy plasma cells or end-stage tissue damage, there are no reliable biomarkers to predict which MGUS patients may develop malignancy and who will remain stable.

Exosomal miRNAs can be easily isolated from almost all the bio-fluids and are interested as a noninvasive way to obtain information about the status of the disease [174]. On the other hand, the high quantity of exosome production in MM patients and their differential miRNA contents in the different stages of myelomagenesis potentiate them as promising biomarkers for early diagnosis, patient prognosis, and monitoring for developing drug resistance (Figure 3).

In a study by Manier et al. on 156 patients with newly diagnosed MM, two circulating exosomal miRNAs, let-7b and miR-18a, were associated with improved survival in patients with MM [175]. Zhang et al. observed that the downregulation of exosomal miRNAs miR-16-5p, miR-15a-5p, miR-20a-5p, and miR-17-5p was correlated with resistance to bortezomib [175]. Furthermore, a meta-analysis study reported that the upregulation of miR-92a and downregulation of miR-16, -25, -744, -15a, let-7e, and -19b are associated with poor prognosis in MM [176].

Although there are plenty of studies on the role of exosomal miRNAs as potential biomarkers in the different stages of MM, there is a lack of consistency in the results. This inconsistency may be attributed to the presence of technical issues in the different array of methods or using platforms. In addition, miRNA profiling in biofluid samples may be affected by the sample size or the technical issues in pre-analytical and analytical steps. Therefore, for confirming the candidate miRNAs as biomarkers for usage in clinical practice, they require to be further validated in larger cohorts.

## 5. Conclusions

The dynamic interactions between tumor cells and the BM microenvironment have a determinant role in the progression of MM. Exosomes regulate bidirectional crosstalks between the different cellular components within the BME. The miRNA cargo in the tumor-derived exosomes is able to reprogram the BME by creating an immunosuppressive environment or enhancement of angiogenesis and by promoting tumor progression. On the other hand, the other cellular components inside the TME contribute to the disease phenotype by shuttling specific exosomal miRNAs to MM cells. Regarding the important role of exosomal miRNAs in the cellular crosstalk during myelomagenesis, improving our knowledge of their function and mechanisms would be important to design effective therapeutic strategies to target TME.

However, translating the preliminary basic studies to clinical applications needs to overcome different challenges. Regarding the complex network of miRNAs and their target genes, selecting the appropriate miRNAs for a therapeutic aim without unknown long-term side effects is the critical challenge. In addition, achieving a safe and effective method for delivering miRNA vectors that ensures their stability is also a problem that needs to be solved before conducting appropriate clinical trials. However, we still need to improve our understanding of the complex interactions in the BM microenvironment during MM development to design the optimal therapeutic/monitoring strategies for MM patients.

## Figures and Tables

**Figure 1 cells-12-01030-f001:**
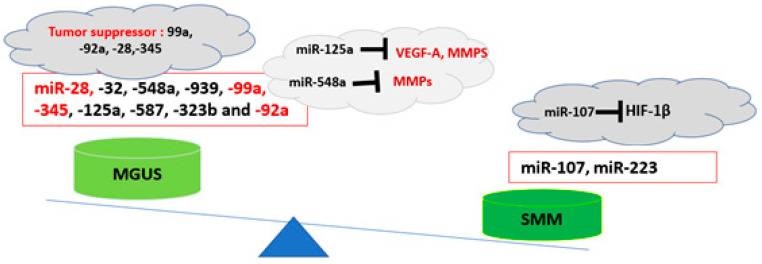
Exosomal miRNAs profile in the pathogenesis of MM. During the progression of MGUS to SMM, the profile of exosomal miRNAs changes towards an immune-suppressive and tumorigenic profile [18].

**Figure 2 cells-12-01030-f002:**
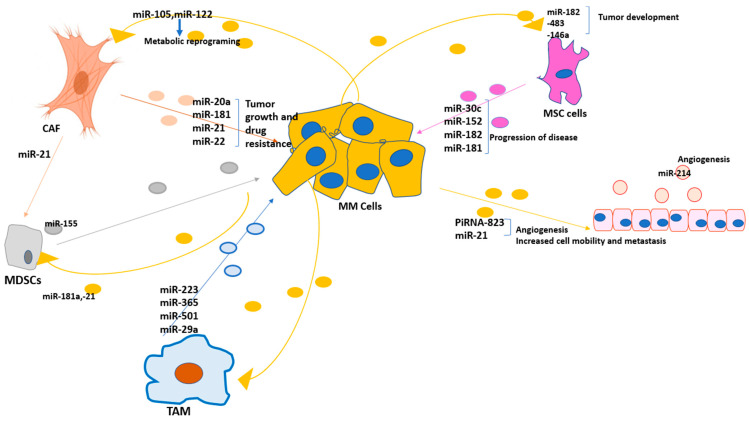
Schematic representation of the potential exosomal miRNA-mediated interactions in the BM microenvironment. Exosomes mediate bidirectional crosstalks between the TME components through shuttling functional molecules, including exosomal miRNAs. MM cells release exosomes with oncomiR cargo that supports tumor progression and/or drug resistance, produces cancer-associated fibroblasts (CAFs), prevents apoptosis, promotes angiogenesis, promotes the expansion of myeloid-derived suppressor cells (MDSCs), and induces an immunosuppressive microenvironment. Other components, in turn, promote the proliferation and survival of tumor cells and support disease progression by the secretion of specific exosomal miRNAs.

**Figure 3 cells-12-01030-f003:**
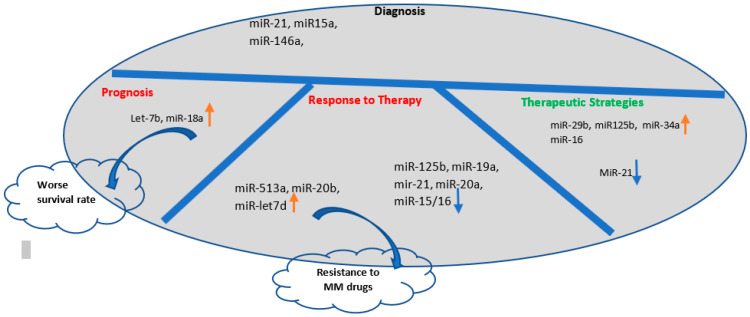
The applications of the exosomal miRNAs in the clinical practice: Exosomal miRNAs are potential diagnostic biomarkers for early detection of MM progress and they are examined for their prognostic potential. In addition, exosomes can be engineered for delivery of therapeutic miRNAs to increase their expression or targeting them to improve the immune response or induction of MM cell apoptosis by reprograming gene expression in the target cells (blue downward arrow shows downregulation and orange upward arrow shows upregulation of miRNAs).

**Table 3 cells-12-01030-t003:** The therapeutic effects of artificial inhibition/overexpression of the miRNAs in MM.

	Therapeutic Cargo	Biological Activities	Key Findings	Refs
1	miR-29b	Targeting the epigenetics modifiers including DNMTs	Synthetic miR-29b mimics improved the aberrant expression of DNMTs in MM cells	[40]
2	miR-15 and-16	Targeting AKT serine/threonine-protein-kinase (AKT3)	Overexpression of miRNA-15a and -16 had showed anti-MM effects (in vitro and in vivo)	[66]
3	miR-324-5p	Targeting Hedgehog (Hh) signaling pathway	Overexpression of miR-324-5p functionally reduced cell growth and cell survival in MM and improved resistance to bortezomib in vitro and in vivo	[162]
4	miR338-3p	Targeting Cyclin-dependent kinases	Overexpression of this miRNA suppressed proliferation and increased the apoptosis of MM cells	[163]
5	miR-152	Targeting DKK1	Over expression of miR-152 improved DR, and inhibited the bone disruption in an intrabone MM mouse model	[67]
8	miR-125b	Targeting IRF4 and BLIMP-1	miR-125b overexpression had an inhibition effect on the proliferation and survival of MM cells and also enhanced apoptosis and cell death in these cells	[164]
9	miR-137/197 synthetic mimics	Targeting MCL-1	Increased the apoptosis and exerted an inhibition effect on the proliferation, colony formation, and migration ability in MM tumor cells	[165]
10	miR-34a mimics	Targeting CDK6, BCL-2, and NOTCH1	Enhanced the apoptosis of MM cells and inhibited the proliferation in these cells	[166]
11	Anti-miR 221/222	Upregulation of PTEN, PUMA, p27Kip1, and p57Kip2.	Induced the antiproliferative effects in MM cells	[167]

## Data Availability

Not applicable.

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
