# Peer review of "Exosomal miRNAs in the Tumor Microenvironment of Multiple Myeloma"

_cells, 2023, doi:10.3390/cells12071030_

Round 1
Reviewer 1 Report
This article is an extensive review of the role of exosomal miRNAs' role in the tumor microenvironment of multiple myeloma. It will help readers understand the current cutting-edge knowledge in myeloma microenvironment and exosomal miRNA. It has 3 figures and 1 table, which are quite well presented to help readers understand.
After minor grammatical errors are corrected, this paper can be published.
Author Response
Thank you for your positive comments. The manuscript has been rechecked for grammatical errors.
Reviewer 2 Report
In this manuscript entitled “Exosomal miRNAs in the tumor microenvironment of Multiple Myeloma”, Alipoor and Chang reviewed recent publications on the roles of exosomal miRNAs in the epigenetic modifications in the tumor microenvironment (TME) and their effects on the developmental phenotypes and prognosis of Multiple Myeloma (MM). MM is one of the most frequently observed hematological malignancies and drug resistance remains a major challenge in their treatment. Regarding the development of drug resistance in MM, crosstalk in TME plays an essential role, in which miRNAs are important cargos that transduce signals among cells in TME. In this review, authors presented the state-of-the-art findings regarding exosomal miRNAs possibly involved in the development of MM and therapeutic strategy for MM focusing on the exosomal RNAs. I believe that this article helps general readers to understand the current status of the basic science regarding MM and its therapy. For general readers, however, miRNA names are somewhat a meaningless row of numbers. Tables listing the exosomal miRNAs help general readers to grasp the whole story of miRNAs.
Comments:
1. Page 4, Line 166: Figure 1: This figure summarizes the role of miRs in the development of MUGS and SMM. It is better to add Table, like Table 1 in Page 12, Line 572, that lists the miRs and other components possibly involved in the development of MUGS and SMM in addition to this figure.
2. Page 9, Line 418: Figure 2: This figure illustrates the potential role of miRs in the development of MM. It is better to add Tables, like Table 1 in Page 12, Line 572, that list the cells and factors such as miRs and other components related to the development of MM in addition to this figure.
Minor comments:
Page 2, Line 62: “tunning” should read “tuning”.
Page 9, Line 428: “abroad” should read “a broad”.
Author Response
Q1: Page 4, Line 166: Figure 1: This figure summarizes the role of miRs in the development of MUGS and SMM. It is better to add Table, like Table 1 in Page 12, Line 572, that lists the miRs and other components possibly involved in the development of MUGS and SMM in addition to this figure.
Response: Thank you for the comment. We added a new table (Table 1) to the manuscript and summarized the information of the profile of miRNA during the transition from MGUS to SMM. “Table 1: The potential role of exosomal miRNAs in the transition from MGUS to SMM”
Q2: Page 9, Line 418: Figure 2: This figure illustrates the potential role of miRs in the development of MM. It is better to add Tables, like Table 1 in Page 12, Line 572, that list the cells and factors such as miRs and other components related to the development of MM in addition to this figure.
Response: Thank you for the comments. We added a new table (Table 2) to summarize the information of the role of miRs in the development of MM. “Table 2: The role of exosomal miRNAs and their cross talk in development of MM”
Q4: Minor comments:
Page 2, Line 62: “tunning” should read “tuning”.
Response: We corrected this word in the revised manuscript.
Page 9, Line 428: “abroad” should read “a broad”.
Response: We corrected this word in the revised manuscript.